# There Is No Spooky Action at a Distance in Quantum Mechanics

**DOI:** 10.3390/e24040560

**Published:** 2022-04-16

**Authors:** Stephen Boughn

**Affiliations:** Departments of Astronomy and Physics, Haverford College, Haverford, PA 19041, USA; sboughn@haverford.edu

**Keywords:** quantum nonlocality, entanglement, wave function collapse, quantum measurement problem, Bell’s theorem

## Abstract

Einstein became bothered by quantum mechanical action at a distance within two years of Schrödinger’s introduction of his eponymous wave equation. If the wave function represents the “real” physical state of a particle, then the measurement of the particle’s position would result in the instantaneous collapse of the wave function to the single, measured position. Such a process seemingly violates not only the Schrödinger equation but also special relativity. Einstein was not alone in this vexation; however, the dilemma eventually faded as physicists concentrated on using the Schrödinger equation to solve a plethora of pressing problems. For the next 30 years, wave function collapse, while occasionally discussed by physicists, was primarily a topic of interest for philosophers. That is, until 1964, when Bell introduced his famous inequality and maintained that its violation proved that quantum mechanics and, by implication, nature herself are nonlocal. Unfortunately, this brought the topic back to mainstream physics, where it has remained and continues to muddy the waters. To be sure, not all physicists are bothered by the apparent nonlocality of quantum mechanics. So where have those who embrace quantum nonlocality gone wrong? I argue that the answer is a gratuitous belief in the ontic nature of the quantum state.

## 1. Introduction

All that follows in this essay is well known to quantum mechanists, although the emphasis is perhaps less familiar. On the other hand, I feel compelled to respond to the frequent references to *spooky action at a distance* which often accompany reports of experiments investigating entangled quantum mechanical states. Most, but not all, of these articles have appeared in the popular press. As an experimentalist, I have great admiration for such experiments and the concomitant advances in quantum information and quantum computing, but, in my opinion, accompanying claims of action at a distance are quite simply nonsense. Some physicists and philosophers of science have bought into the story by promoting the nonlocal nature of quantum mechanics. In 1964, John Bell [1] proved that classical hidden variable theories cannot reproduce the predictions of quantum mechanics unless they employ some type of action at a distance. I have no problem with this conclusion. Unfortunately, Bell later expanded his analysis and deduced that quantum mechanics and by implication nature herself are nonlocal.

In addition, some of these articles present Einstein in caricature, a tragic figure who neither understood quantum mechanics nor believed it to be an accurate theory of nature. Consequently, the current experiments have proven him wrong. This is also nonsense. Einstein [2] believed, “…that quantum mechanics has seized hold of a beautiful element of truth, and that it will be a test stone for any future theoretical basis…” He simply did not like the theory and felt “…that it must be deducible as a limiting case from that [future theoretical] basis…” I am sure he would have accepted the current experimental findings on entangled states and concluded that they confirm the spooky action at a distance, which he thought quantum mechanics implied. In this sense, he and those who accept the nonlocality of quantum mechanics are in complete agreement. Their disagreement is more metaphysical than physical. It is simply that while the latter group are content with this situation, Einstein was not. As I will explain in what follows, I find both of these positions unsatisfactory (The book by Douglas Stone [3] documents Einstein’s extensive contributions to quantum mechanics and the paper by Don Howard [4] similarly corrects myths about the Einstein–Bohr debates).

To be sure, there are other contexts in which the possibility of nonlocality has been raised. They are discussed in some detail in the book by George Musser entitled *Spooky Action at a Distance* [5]. Among them are the black hole information paradox, space-time wormholes, cosmological inflation, and the quantum structure of space-time itself. These are highly theoretical speculations and, because of my experimentalist proclivities, I will refrain from discussing them further and confine my comments to experimentally confirmed predictions of standard quantum mechanics.

## 2. Einstein’s Spooky Action at a Distance

As far as I know, Einstein first used the phrase “spooky actions at a distance” in a 1947 letter to Max Born [6]. I wish he had not. It is such a catchy phrase; who can resist it? While the 1935 paper by Einstein, Podolsky, and Rosen (EPR) [7] is often credited with bringing the notion of entangled quantum states and action at a distance to the attention of the physics community, the major figures in physics at the time were already arguing about entanglement and Einstein had been bothered by action at a distance since the 1927 Solvay conference [4]. At Solvay, he used a single-slit gedanken experiment to illustrate the action at a distance required by quantum mechanics. Consider the simple case of diffraction by a small hole. The Schrödinger wave function of a particle emanating from the hole is essentially a spherical wave indicating that the particle can be detected at any point on a hemispherical screen with equal probability. Therefore, the detection of the particle at any point on the screen must cause the wave function to instantaneously vanish everywhere except at that point, hence, action at a distance. He pointed out that this was not a problem if the wave function is taken to represent the behavior of an ensemble of particles; however, if it is to provide an objective description of a single particle, then it clearly violates special relativity [4]. Einstein was not challenging the veracity of the predictions of quantum mechanics but rather its ontological interpretation.

I first read the EPR paper when I was in graduate school. I was mystified. The paper seemed to be less about physics and more about the philosophical issue of what constitutes reality. To be sure, the conclusion of EPR was not that quantum mechanics is nonlocal nor that objective reality does not exist but rather that quantum mechanics “does not provide a complete description of the physical reality”. In fact, the paper was penned by Poldolsky and Einstein was not happy with it. Howard [4] points out that in a letter to Schrödinger written a month after the EPR paper was published, Einstein chose to base his argument for incompleteness on what he termed the “separation principle” and continued to present this argument “in virtually all subsequent published and unpublished discussions of the problem”. According to the separation principle, the *real state of affairs* in one part of space cannot be affected instantaneously or superluminally by events in a distant part of space. Suppose *AB* is the joint state of two systems, *A* and *B*, that interact and subsequently move away from each other to different locations. In his letter to Schrödinger, Einstein explained [4].
After the collision, the real state of (*AB*) consists precisely of the real state *A* and the real state of *B*, which two states have nothing to do with one another. *The real state of B thus cannot depend upon the kind of measurement I carry out on A* [separation principle]. But then for the same state of *B* there are two (in general arbitrarily many) equally justified [wave functions] $\Psi_{B}$, which contradicts the hypothesis of a one-to-one or complete description of the real states.

His conclusion was that quantum mechanics cannot provide a complete description of reality. Note that Einstein’s separation principle did not claim that a measurement of system *A* has no effect on the *result* of any measurement on system *B* but rather that a measurement of system *A* has no effect on the *real state* of system *B*. This distinction will be elaborated on in Section 4.

## 3. Entanglement

While the instantaneous change of the single-particle wave function discussed above already manifests action at a distance, it is the joint state of two previously interacting particles and two separate observers (also discussed above) that clearly leads to the notion of nonlocality. The EPR paper introduced such a joint state, which Schrödinger [8] later christened *entangled*. Perhaps the quintessential example of an entangled state is the one introduced by Bohm and Aharanov [9] and subsequently used by Bell in his 1964 paper. It consists of the emission of two oppositely moving spin ½ particles in a singlet state, i.e., the spins of the two particles are precisely opposite each other but in an indeterminate direction. According to quantum mechanics, their combined wave function (quantum state) is given by
(1)$$\Psi \left(1,2\right)=\frac{1}{\sqrt{2}}\{|1,↑\rangle_{z}|2,↓\rangle_{z}-|1,↓\rangle_{z}|2,↑\rangle_{z}\}$$
where ↑ and ↓ indicate the up and down *z* components of the spins of particles 1 and 2. Now suppose that the spin of particle 1 is measured with a Stern–Gerlach apparatus oriented in the $\hat{z}$ direction and is determined to be ↑. Then, we know the *z* component of the spin of particle 2 will be ↓. That is, particle 2 is in the state $\Psi \left(2\right)=|2,↓\rangle_{z}$. This perfect (anti)correlation is built into the two–particle system because they are in a singlet state. On the other hand, the original wave function can also be expressed in terms of the *x* components (or in any direction for that matter) of the spin,
$$\Psi \left(1,2\right)=\frac{1}{\sqrt{2}}\{|1,↑\rangle_{x}|2,↓\rangle_{x}-|1,↓\rangle_{x}|2,↑\rangle_{x}\}$$

Then, if the spin of particle 1 is measured with a Stern–Gerlach apparatus oriented in the $\hat{x}$ direction and is determined to be ↑, the spin of particle 2 will be ↓, i.e., particle 2 is in the state $\Psi \left(2\right)=|2,↓\rangle_{x}$, a state fundamentally distinct from $|2,↓\rangle_{z}$. The immediate transition of particle 2 to either the state $|2,↓\rangle_{z}$ or $|2,↓\rangle_{x}$, depending on the measurement performed on the distant particle 1, is often referred to as “state reduction” or “collapse of the wave function” and is the basis for claims of “action at a distance”. In their venerable quantum mechanics texts, Dirac [10] and von Neumann [11] both gave detailed treatments of state reduction although it is not clear to what extent they subscribed to the notion of action at a distance. After Bell’s papers, however, some physicists and philosophers embraced the nonlocality inherent in quantum mechanics and the concomitant notion of instantaneous action at a distance. Likewise, Einstein felt that quantum mechanics implied action at a distance and, therefore, violated his separation hypothesis. Consequently, he concluded that quantum mechanics failed to provide “a one-to-one or complete description of the real states”; ergo, quantum mechanics is incomplete. So Einstein and adherents of quantum nonlocality both subscribe to spooky action at a distance but, while the latter embrace it, Einstein inferred from it that quantum mechanics does not provide a complete description of physical reality.

## 4. The Pragmatic Perspective

Now consider a more pragmatic perspective that is essentially Bohr’s Copenhagen interpretation. For the entangled singlet state considered above, Einstein would conclude particle 2 has no unique (real) state. Let me push back on this conclusion. After their emission, the polarization of neither particle is known. Therefore, the two particles can be considered as two unpolarized particle beams. For particle 2, this can be represented by a 50% mixture of spin up, |2,↑〉, and spin down, |2,↓〉, states (in any direction), a so-called *mixed state*. A mixed state cannot be described by a pure quantum state (e.g., the Schrödinger wave function or vector in Hilbert space) but is well described by its associated density matrix ϱ. The density matrix for the unpolarized spin ½ particle 2 is given by
(2)$$\rho \left(2\right)=\frac{1}{2}|2,↑\rangle \langle 2,↑|+\frac{1}{2}|2,↓\rangle \langle 2,↓|$$
with a similar expression for particle 1. This density matrix is sufficient to completely describe any possible (spin) measurement of particle 2. For example, the expectation value (mean measurement) of any observable represented by an operator *A* is given by ⟨A〉=tr(ρA), i.e., the trace of the matrix ρA and, as always, the value of a single measurement is one of the eigenvalues of *A*. For the above expression, this implies that a measurement of the spin component in any direction will yield ↑ for 50% of the measurements and ↓ for 50% of the measurements subject to the usual statistical fluctuations. This completely characterizes the spin measurements of either particle with no mention of what measurements are made on the other particle. That is, the *result* of the measurement made on particle 2 is completely independent of any measurement carried out on particle 1 (After all, the measurement made on particle 2 could be made in advance of even selecting the type of measurement to be made on particle 1.). I would call this the experimentalist’s or pragmatist’s separation hypothesis in contrast to Einstein’s version that the *real state* of particle 2 must be independent of any measurement carried out on particle 1. I know of no observation or prediction of standard quantum mechanics that violates the pragmatist’s separation hypothesis (I have discussed this topic in greater detail in *Making Sense of Bell’s Theorem and Quantum Nonlocality* [12]). As for Einstein’s separation hypothesis, a pragmatist would probably object to the notion of a *real state* or at least defer comment until a precise definition of “real state” is offered. I will come back to the notion of the reality of quantum states in Section 5 and Section 6.

One might object that Equation (2) was pulled out of thin air and suggest that a proper quantum mechanical treatment might provide a more complete description of the individual particle states. Not so. Consider the density matrix of the pure entangled state in Equation (1). It is given by ρ(1,2)=|Ψ(1,2〉⟨Ψ(1,2)| and is in every way equivalent to the entangled state Ψ(1,2) itself. So how do we conjure a single-particle 2 state from the entangled state? The standard method is to marginalize the two-particle density matrix over the possible states of particle 1. This is accomplished by taking the *partial trace* of the density matrix ρ(1,2) over the basis of system 1, i.e.,
ρ(2)=⟨1,↑| ρ(1,2)|1,↑〉+⟨1,↓| ρ(1,2)|1,↓〉

This is precisely the density matrix of Equation (2). That is, Equation (2) describes the complete quantum state of particle 2 and characterizes any measurement made on that particle, of course subject to the statistical nature of all quantum mechanical predictions.

Okay, so far so good. Nevertheless, you may object that these single-particle mixed states are mute on the correlations of measurements made on the two states. Fair enough, but now you are asking a different question. That is, after making these independent measurements, what are the correlations between them? To answer this question, to be sure, one needs the pure entangled state of Equation (1) but this expression in no way implies that the measurements made on particle 1 have any effect whatsoever on the measurements made on particle 2. One might exhort, as did John Bell [13], “The scientific attitude is that correlations cry out for explanation.” Well, maybe but as we often have to remind our students, correlation does not necessarily imply causation and in any case one can certainly trace the source of the correlation back to the interaction between the two particles prior to their emission. There is a simple classical analog that emphasizes this point. Suppose one randomly places either a white ball or a black ball in a box and then places the remaining ball, which has the opposite color, in another box. Now the two boxes are closed and sent off in opposite directions where they will encounter observers who open the boxes. It is clear that each observer has a 50/50 chance of finding a white ball and a 50/50 chance of finding a black ball and that completely characterizes the state of the unopened box for each individual observer. This can be confirmed by compiling the data from multiple (random) trials of the experiment. Yet it is also absolutely clear that the findings of the two observers will be completely correlated; if the observer of box 1 finds a white ball, the observer of box 2 will necessarily find a black ball and vice versa. However, we would never claim that the act of observing a white ball in box 1 *causes* a black ball to appear in box 2. Of course, quantum entanglement is a much richer phenomenon than classical entanglement as the above spin singlet entangled pair illustrates. Whereas in the classical case it can be argued that each of pair of balls has a definite, but unknown color, the same cannot be said in the quantum state for which neither of the two particles has a definite spin, the situation that so vexed Einstein. This arises from standard quantum mechanical superposition, e.g., Equation (2) and the following equation, the principle that lies at the heart of quantum interference and wave/particle duality. It has little to do with the notion of entanglement and certainly does not provide evidence of nonlocality and action at a distance.

Still, after we observe particle 1 and determine the *z* component of its spin to be ↑, are we not justified in identifying the state of particle 2 to be $\Psi \left(2\right)=|2,↓\rangle_{z}$? In which case, did we not cause its quantum state to collapse from its original mixed state in Equation (2) to $|2,↓\rangle_{z}$? On the other hand, if we decided to observe particle 1 in the x direction and found it to be ↑, then did we not cause the state of particle 2 to be $\Psi \left(2\right)=|2,↓\rangle_{x}$? It sure seems like some kind of nonlocality is afoot. How else might one *explain* the correlations of the states of the particles? To answer these questions, it is necessary to look more closely at the meaning of the term ‘quantum state’ or ‘quantum wave function’.

## 5. The Quantum Wave Function

So just what is the meaning of a quantum wave function? By “meaning”, I am referring to the empirical meaning that any experimentalist like me needs to know. Stapp’s 1972 paper [14] on the Copenhagen interpretation is particularly lucid. In Stapp’s practical account of quantum theory, a system to be measured is first prepared according to a set of specifications, *A*, which are then transcribed into a wave function $\Psi_{A}\left(x\right)$, where *x* are the degrees of freedom of the system. The specifications *A* are “couched in a language that is meaningful to an engineer or laboratory technician”, i.e., not in the language of quantum (or even classical) formalism. Likewise, *B* are a set of specifications of the subsequent measurement and its possible results. These are transcribed into another wave function $\Psi_{B}\left(y\right)$, where *y* are the degrees of freedom of the measured system. How are the mappings of *A* and *B* to $\Psi_{A}\left(x\right)$ and $\Psi_{B}\left(y\right)$ affected? According to Stapp [14],
…no one has yet made a qualitatively accurate theoretical description of a measuring device. Thus what experimentalists do, in practice, is to *calibrate* their devices…[then] with plausible assumptions…it is possible to build up a catalog of correspondences between what experimentalists do and see, and the wave functions of the prepared and measured systems. It is this body of accumulated empirical knowledge that bridges the gap between the operational specifications *A* and *B* and their mathematical images $\Psi_{A}$ and $\Psi_{B}$. Next a transition function U(x;y) is constructed in accordance with certain theoretical rules…the ‘transition amplitude’ $\langle A|B\rangle \equiv \int \Psi_{A}\left(x\right)U\left(x;y\right)\Psi_{B}^{*}\;dxdy$ is computed. The predicted probability that a measurement performed in the manner specified by *B* will yield a result specified by *B*, if the preparation is performed in the manner specified by *A*, is given by $P\left(A,B\right)={\left|\langle A|B\rangle \right|}^{2}$.

What is my point here? It is that the quantum mechanical wave function is a theoretical construct that we invented to deal with our observations of physical phenomena. As such, it seems reasonable that we derive our understanding of it according to how we use the concept. Stapp’s (and Bohr’s) pragmatic account of wave functions is intimately tied to state preparation and measurement, both of which are described in terms of operational specifications that lie wholly outside the formalism of quantum mechanics. Prior to the preparation of a system, the wave function is not even defined and after the measurement the wave function ceases to have a referent. To extrapolate this notion of the wave function to being a fundamental building block of nature is an enormous leap that necessitates dropping all references to the operational specifications that gave wave functions their meanings in the first place. One might be lead to such an extrapolation by ascribing an ontic reality to the notion of wave function (despite its epistemic transitory nature). This is especially tempting when presented with some of the incredible successes of quantum theory. I was certainly enticed to do so when I first learned of the twelve-decimal-place agreement of the quantum electrodynamic prediction with the measured value of the g-factor of the electron (When I was a graduate student, it was considered prescient that the acronym for quantum electrodynamics (QED) was the same as for the Latin phrase quod erat demonstrandum).

So after the measurement of particle 1, what is the quantum state of particle 2? Is it still the density matrix ρ(2) of Equation (2) or is it the wave function $|2,↓\rangle_{z}$? The premise of this question is wrong. The quantum state is not an “it” in this sense. The only “it” is the quantum system, the spin ½ particle. The quantum state is not a physical object but rather a description of a physical object, a transcription arising from “a set of specifications”. For the physicist who has just observed particle 1 to be in the ↑ state, the set of specifications leads to $\Psi \left(2\right)=|2,↓\rangle_{z}$ while for the future observer of particle 2, the available set of specifications leads to the density matrix ρ(2) of Equation (2). If the result of the observation of particle 1 is communicated (subluminally, of course) to the observer of particle 2, then the Copenhagen interpretation would simply point out that the original specifications of the quantum system must be replaced by new ones that include the result of that measurement. “Then the original wave function [or density matrix in this case] will be naturally replaced by a new one, just as it would be in classical statistical theory.” [14]. Some physicists prefer to claim, euphemistically, that wave function collapse happens in the mind of the observer but not for any physical reason. For example, Hartle [15] explains.

The “reduction of the wave packet” does take place in the consciousness of the observer, not because of any unique physical process which takes place there, but only because the state is a construct of the observer and not an objective property of the physical system.

## 6. Final Remarks

I opened this essay with Einstein’s contention that quantum mechanics implies “spooky action at a distance” and that physicists and philosophers who accept quantum nonlocality are in agreement. Their disagreement is a metaphysical one. Einstein did not like this aspect of quantum mechanics, whereas the latter group embrace it. So where do I think they go wrong? Even though in my youth I was mystified by the EPR paper, I think it points to the fundamental problem, a belief in the ontic nature of the quantum state. As I alluded to above, there is often a tendency to identify theoretical constructs of highly successful models with reality itself. One can argue that this is as much the case with classical mechanics as with quantum mechanics. The “particle” construct of the former is considered to be real in the same sense as the “wave function” construct of the latter. Once one is lead down this path, it is inevitable to conclude that spooky action at a distance occurs in nature. On the other hand, if one eschews the ontological interpretation of the wave function, then “action at a distance” is, at best, descriptive of the mathematical formalism of quantum mechanics. One must be extremely wary of extending such inferences to physical reality itself. I have always thought that the proofs that standard quantum mechanics forbids sending superluminal signals should have disabused nonlocality advocates of their claims of action at a distance. However, once one attaches ontic significance to theoretical constructs, such as the wave function, such metaphysical conclusions are inevitable.

I just indicated that the disagreement of Einstein with quantum nonlocality advocates is a metaphysical one. On the other hand, I have to admit that the disagreement between those of us who disparage quantum nonlocality and those who do not is also, in a sense, metaphysical and I doubt that there will ever be complete agreement on the dispute. Why do I say this? Because the physicists in both of these camps consider quantum mechanics to be a powerful theory and, as far as I know, have absolute agreement on all quantum mechanical predictions. I hasten to add that it is also my impression that the majority of physicists are in neither of these camps but rather are content with viewing quantum mechanics in terms of the incredible usefulness of the theory. So why do I criticize the notion of quantum nonlocality? Primarily, because I am a pragmatist and do not find the concept of quantum nonlocality at all useful. I also worry that it might be a distraction that diverts attention from more productive efforts.

You have probably noticed that I cited none of the supposedly frequent popular articles that reference the notion of “spooky action at a distance” nor any of the many professional physicists and philosophers who have written about the philosophical foundations of quantum mechanics and who fall on both sides of the quantum nonlocality dispute. I simply assume that the reader will have already encountered one or more of them. On the other hand, I doubt those who have not or who are not particularly interested in the philosophical foundations of quantum mechanics will be motivated to read this essay. Additionally, I do not mean to imply that all popular articles on entangled states support the action at a distance hypothesis. In 2016, Tom Siegfried [16], past editor of *Science News*, published in that magazine a pair of articles, the first entitled “Entanglement is spooky, but not action at a distance”, which disabuses one of the notion that entanglement implies action at a distance. However, Siegfried still finds entanglement “spooky” in a way I do not. He also implies that Einstein might have changed his mind had he lived long enough to read Bell’s papers and know of the subsequent experiments on entangled states. I suspect not. On the contrary, I can imagine that Einstein would have agreed with everything I have written in this essay, just as he proclaimed the “beautiful element of truth” of quantum mechanics. However, he would continue to maintain that it does not provide a precise, or even approximate, description of physical reality. In a sense, I agree, except that from my experimentalist’s pragmatic vantage point, “physical reality” is something I accept and not something that I aspire to accurately portray.

## Data Availability

Not applicable.

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
