# Peer review of "There Is No Spooky Action at a Distance in Quantum Mechanics"

_entropy, 2022, doi:10.3390/e24040560_

Round 1
Reviewer 1 Report
It’s always a pleasure to read reflections on quantum mechanics from an experimentalist. Just as there’s much to be learned from theorists who’ve spent their career in quantum foundations, there’s something equally valuable to be learned from those who’ve spent their time in the lab, and who are now reflecting on the nature of their practice and what it says about the theory. “There Is No Spooky Action at a Distance in Quantum Mechanics,” was a delightful read, clearly written and concise, readable by the expert and the canny layperson alike. It provides just enough historical context to motivate the philosophical issue in a satisfying and edifying way, and just enough mathematics to give it heft. It’s always refreshing for quantum mechanics to be returned to its roots, and to hear the language of Bohr recalled from the past. Moreover, the reification of the quantum state is an ongoing issue both in popular culture (with its love for multiverses) and in academic literature, and so this essay is as timely as ever.
Indeed, the basic contention of this piece is that the apparent “spookiness” of quantum mechanics results from a conceptual error, whereby the the quantum state is divorced from the context of its actual use and projected onto reality as if it were an image of the things themselves, rather than a tool for bridging experimental preparations and measurements. I found the section based on Stapp’s work to be particularly good. I also appreciated the author’s critique of the popular notion of Einstein as tragic figure, unable to accept or understand quantum mechanics, as well as its dive into the history of the term “spooky action at a distance,” emphasizing that Einstein wasn’t challenging quantum mechanics per se, but its ontological interpretation.
One comment I have, however, is that for the general reader’s sake, it might be worth spelling out in more detail the relationship between the “spooky action at a distance” in the single particle case, whereby the detection of a particle’s location apparently causes the wavefunction to vanish everywhere else, and the supposed “spooky action at a distance” associated with two entangled systems. On that note, I quite liked the distinction made between Einstein’s separation principle, which holds that a measurement on some system A has no effect on the “real state” of system B, and the so-called pragmatist’s separation principle, whereby merely the result of a measurement made on system B is completely independent of a measurement carried out out system A. This is taken to be formalized in the assignment of maximally mixed density matrices to the partial states of, for example, two spins in a singlet, by which the result of any measurement is predicted to be 50/50. The author then writes, “Nevertheless, you may object that these single particle mixed states are mute on the correlations of measurements made on the two states. Fair enough, but now you are asking a different question. That is, after making these independent measurements, what are the correlations between them?… One can certainly trace the source of the correlation back to the interaction between the two particles prior to their emission.”
It may, however, be worth emphasizing at this juncture that unlike in the classical case, the correlation can’t be because of a prior agreement regarding some classical local hidden variables, especially to further dramatize the problems associated to Einstein’s notion of the “real state.” Indeed, the discussion then segues into a comparison with classical correlations between black and white balls. (Incidentally, there is a minor omission here: “Suppose one randomly places either a white ball or a black ball in a box and then places the remaining ball in another box.” It should say something like, “and then places a second ball, which has the opposite color.”) After this, the author avers: “Of course, quantum entanglement is a much richer phenomenon than classical entanglement and exhibits all the aspects of quantum interference with which we are familiar. But this…certainly does not provide evidence of nonlocality and action at a distance.” I feel that this is perhaps too deflationary, and unless the reader is already well versed in quantum theory, it would be difficult to pin down from this discussion exactly in what sense quantum entanglement is “richer,” and why anyone would think it non-classical at all.
Indeed, while “spooky” might be too overloaded a term, it is somewhat surprising that, if (in many runs of an experiment) I measure one half of a singlet along some direction, and then get a report from someone who measured the other half (both the result and the direction they measured in), that I should expect their results to be consistent with their spin having been oriented in the opposite direction to my own, precisely because the direction I measured in was my own free choice and couldn’t have been known to the spins at the time of their “agreement.” Presented with this notion, it isn’t obviously absurd that some kind of nonlocality is afoot, and I’d say it’s only upon reflection that one comes to the view that this can be explained in completely local terms, especially upon the consideration that it’s precisely the randomness of the individual outcomes that prevents these correlations from allowing one to communicate superluminally. (This is alluded to in Section 6, but might serve to be mentioned here at the end of Section 4.)
In other words, while the classical analogy is helpful, it doesn’t quite do justice to quandary: collapse or no, why should one expect the results of the two measurements to be correlated (even just within one observer’s experience) in just the case where there isn’t a local hidden variable theory? In discussing articles in the popular press, the author writes, “…but the accompanying claims of action at a distance are quite simply nonsense. Some physicists and philosophers of sciences have bought into the story by promoting the nonlocal nature of quantum mechanics.” Are we to understand the latter to be acting not just under a conceptual error, but also in bad faith? I would hate for the reader to come away from this piece with a cynicism about the state of physics. And this connects up to my larger point that it might be worth making the case for nonlocality a little stronger, or at least to go a little deeper into the reasons why so many smart people have signed onto it, if only to have a more satisfying victory over it in the end.
In fact, I would contend that the desire to reify the quantum state may not just come from a natural conflation of a remarkably predictive model with reality, but also out of the desire to avoid what I would call subjective/relational/participatory views of quantum mechanics. For example, the Many Worlds Interpretation and its cousins are good examples of ontological interpretations that are explicitly local, showing that at least for some, these are two separate issues. At the other extreme, QBism holds explicitly that quantum mechanical predictions are predictions about one’s own future experiences, and thus the theory is manifestly local, but its emphasis on subjectivity often rubs people the wrong way (although not me!).
Now, it is perfectly reasonable to take a purely pragmatic point of view, and avoid philosophical commitments on the nature of subjectivity and objectivity, and nothing could be more charming than the quote with which the author ends the essay (“from my experimentalist’s pragmatic vantage point, ‘physical reality’ is something I accept and not something that I aspire to accurately portray”). At the same time, it might be beneficial to point the reader towards some contemporary work in quantum foundations which are generally in harmony with the author’s views, if only to show that, although some of the voices acclaiming nonlocality are quite loud, they don’t necessarily represent even the majority view, and you don’t have to go so far back as Stapp to find the spirit of Copenhagen alive and well. Then again, it wouldn’t be appropriate to enter into all the fine details of the zoo of interpretations, and the paper is served well by its basic self-containedness.
Author Response
I very much appreciate the extensive comments of the reviewer who clearly thoroughly perused my paper. I made several (minor) changes to the manuscript that addressed most (but not all) of the reviewer's comments. Here is a brief list of the changes:
1) I added a few words in section 3 to differentiate single particle action at a distance and entangled particle nonlocality.
2) In section 4, I corrected an oversight by adding the requirement that the two balls are of “opposite” colors.
I also added a few words to differentiate the classical (hidden variable) case with the quantum case and tried to indicate just why it is that quantum entanglement is richer than classical entanglement. Of course, a detailed discussion of this issue would require a considerable diversion from the topic at hand.
I also added at the end of the section, in the reviewer’s words, that it sure seems like “some kind of nonlocality is afoot” but refrained from claiming that “it’s only upon reflection that one comes to the view that this can be explained in completely local terms, especially upon the consideration that it’s precisely the randomness of the individual outcomes that prevents these correlations from allowing one to communicate superluminally.” Quantum mechanics is local and is inherently random but explaining the former in terms of the latter, at least for me, isn’t very enlightening.
3) Regarding my cheeky comment about action at a distance being “nonsense”: I deliberately penned my essay in a contrarian tone to counteract the many popular press claims that nature is nonlocal. In order to soften my criticism, I made it clear in section 6 (and in the abstract) that this was simply my view. I also added some words in the this section that renders my position less dogmatic. I personally don’t think quantum nonlocality is “incorrect” nor will there ever be complete agreement about this metaphysical issue. The fact is that all parties involved believe in the accuracy of quantum mechanics and rarely, if ever, disagree about quantum mechanical predictions. The dilemma is more philosophical (and hence personal) that physical. I hope my addition will be helpful and yet won’t totally undercut my argument. :)
Related to the previous comment, I’ve resisted going more deeply into the variety of views on quantum nonlocality that would involve delving into the many, many interpretations of quantum mechanics. That would be a momumental task and one that I’m not particularly qualified to undertake. I did make note in section 6 of the many physicists that lie on both sides of the issue as well as pointing out the larger majority who have little interest in it.
Reviewer 2 Report
The material is presented by the author in a form of an essay which basically discusses the so called “spooky action at a distance” in many-body quantum theory which reflects the effect of a measurement of certain part of the quantum system on the rest of it. I think the paper cannot be considered strictly a research paper but I also think it could be appropriate for a Special issue on the completeness of quantum theory where more philosophically oriented work would be plausible. Since this is still a hot topic of research I would recommend the paper for publication in Entropy.
Author Response
I agree with the reviewer that my essay is more philosophical than physical and have added a few words in the text to that effedt.
Reviewer 3 Report
The ideas in the paper are discussed very widely and many of their remarks may be questioned. Nevertheless I think overall the paper is worthy to be published.
Author Response
I agree with the implication of the reviewer that nearly all remarks made by me and by others may be questioned, as is usually the case in more philosophically oriented discussions.